# ENHANCING THE REPRESENTATIONAL POWER OF SPIKING NEURAL NETWORKS VIA DIVERSIFIED SPIKE PATTERNS

## ABSTRACT

One of the fundamental aspects of spiking neural networks (SNNs) is how they encode and process information through the generation of spikes, and direct coding is one of the most widely used coding schemes for its simplicity and promising performance. In this study, we examine the traits of the encoded spike trains under the direct coding scheme and reveal that the severe imbalance in the distribution of spike train patterns can pose a major obstacle to SNN performance. Based on our analyses, we propose diverse-pattern coding (DPC), a novel neural coding scheme that diversifies encoder output spike patterns through two technical components: temporal embedding and temporal feedback layer. The former incorporates information over time into the input, and the latter applies a recurrent layer for each timestep to deliver heterogeneous features to the input spiking neuron. Our extensive experimental results demonstrate that DPC improves SNN performance through diversified encoded spikes, achieving superior performance across multiple datasets and model architectures with minimal increase in memory costs.

## 1 INSTRUCTION

Spiking neural networks (SNNs), a promising next generation of artificial neural networks (ANNs), are inspired by the event-driven computation of human brains (Maass, 1997). They have garnered attention for their high efficiency, bio-plausibility, and low power consumption on neuromorphic hardware such as TrueNorth (Merolla et al., 2014), Loihi2 (Orchard et al., 2021), and NorthPole (Modha et al., 2023). Unlike ANNs, which transmit information using floating-point values, SNNs convey information efficiently through binary spikes with temporal dynamics. To address the mismatch between inputs and the temporal nature of SNNs, static data must be transformed into spatio-temporal spike trains by introducing a time dimension and binarizing real-valued inputs (Eshraghian et al., 2023). This process, known as neural coding, serves as a bridge between static real-valued input and binary spike trains and is essential for enabling SNNs to perform their tasks.

Neural coding schemes can be distinguished based on how they convert real-valued signals into spike trains; most widely-adopted methods include rate coding (Van Rullen & Thorpe, 2001), temporal coding (Zhou et al., 2021), and direct coding (Rathi & Roy, 2021; Wu et al., 2019). Rate coding generates spikes proportional to the input value. Temporal coding, such as TTFS coding (Park et al., 2020), encodes information at the time of spike firing. However, due to the gradient vanishing problem in deep neural networks, temporal coding has not been broadly used in deep SNNs (Eshraghian et al., 2023; Shrestha & Orchard, 2018; Zheng et al., 2018). Direct coding, as the name implies, involves transmitting the real-valued features produced by the encoding layer *directly* to the first spiking neuron, and this allows direct coding to deliver better performance in various deep architectures (Rathi & Roy, 2021). Due to its promising results and simplicity of implementation, direct coding has been unquestionably chosen as a standard approach across various studies without further scrutiny (Hu et al., 2024; Yao et al., 2023).

As mentioned above, direct coding feeds the same input value at every timestep during forward propagation. Our thorough analyses demonstrate that repeated inputs cause a severe imbalance in spike train patterns, subsequently limiting their diversity. We use the entropy of spike train pattern distribution as a metric to quantify the diversity of spike trains and conduct an empirical study to

assess its impact on model performance. The results show that this intrinsic lack of spike train diversity under a direct coding scheme constrains the performance of SNNs.

To address this problem, we propose diverse-pattern coding (DPC), a novel neural coding scheme that improves the representational power of SNNs by diversifying the encoded spike trains. DPC introduces two novel technical components to improve the diversity of spike trains by integrating time-varying information into the encoded spikes. First, a *temporal embedding* that integrates learnable temporal information into the input space is used to remove the repetitiveness of static inputs. We also utilize a *temporal feedback layer*, which leverages a recurrent connection that projects spikes from the previous timestep to the features of the current timestep. Previous works that aim to address the limitation of direct coding have targeted specific model structures (*e.g.,* modifying the attention mechanism within Transformers), which makes them inherently incompatible with other architectures (Shen et al., 2024; Zhu et al., 2024). In contrast, our method is a general approach that can easily be integrated in a plug-and-play manner into the neural coding stage of various architectures.

To evaluate the effectiveness of our DPC, we conduct extensive experiments across diverse tasks, including static/neuromorphic image classification, time series forecasting, and natural language understanding. We also verify the general applicability of DPC on various model architectures, such as ResNets and Transformers. The results clearly demonstrate that incorporating DPC consistently improves SNN performance across diverse data types and architectures by mitigating the imbalance in encoded spike trains, underscoring its robustness and broad generalizability. Efficiency analysis shows that DPC introduces only a marginal increase in model parameters and energy consumption compared to direct coding. In addition, component ablation demonstrates how each element of DPC contributes to spike-train diversity, and a spike-shuffling test confirms that DPC encodes richer temporal information than direct coding. Our main contributions can be summarized as follows:

- We investigate the pattern diversity of encoded spikes under a direct coding scheme in SNN models and empirically show the correlation between diversity and performance with spike train entropy.
- We propose diverse-pattern coding (DPC), a simple yet effective method that addresses the shortcomings of direct coding. By integrating temporal information into encoded spike trains, our approach enhances their pattern diversity.
- We demonstrate the effectiveness of the proposed method across various datasets and model architectures. By substituting direct coding with DPC, our method not only surpasses previous state-of-the-art approaches but also maintains efficiency across diverse timesteps.

## 2 RELATED WORKS

**Neural coding schemes for SNNs**   One of the fundamental aspects of SNNs is how they encode and process information through discrete spikes that represent neuronal activity over time. Neural coding schemes play a key role in translating input stimuli into spike patterns (Auge et al., 2021; Chen et al., 2024; Kim et al., 2022). Rate coding (Van Rullen & Thorpe, 2001) is one of the simplest coding schemes, where information is encoded by the average firing rate of neurons over a time window. In rate coding, higher input magnitudes lead to higher firing rates. While straightforward to implement, rate coding requires long timesteps to reduce quantization error (Tavanaei et al., 2019). Temporal coding, in contrast, utilizes the precise timing of spikes, such as time-to-first spike and inter-spike intervals, to convey information (Park et al., 2020; Thorpe et al., 2001). Though more expressive, it is sensitive to noise and often requires longer timesteps, making it difficult to use in complex architectures (Eshraghian et al., 2023). Phase coding involves encoding information in the relative phase of spikes with respect to a global oscillatory signal, allowing for a more efficient and precise representation of temporal patterns (Kim et al., 2018). Burst coding is an encoding strategy where information is conveyed through rapid sequences of spikes, known as bursts, from individual neurons (Park et al., 2019). These bursts consist of multiple spikes occurring in quick succession within a short time window. Direct coding represents a significant advancement over traditional methods by enabling more efficient information encoding (Rathi & Roy, 2021; Wu et al., 2019). It works by feeding the inputs through an ANN encoding layer first to produce real-valued features. The features are then passed to spiking neurons, which generate spikes. This reduces the need for extensive spike trains and enables faster, more accurate decision-making (Kim et al., 2022). Such efficiency is crucial for tasks requiring low latency and high accuracy, which is why direct coding

has been widely adopted in many state-of-the-art SNN applications. Nevertheless, recent work has pointed out that direct coding generates periodic spike trains through the repeated injection of the same input values and criticized them as being powerless (Qiu et al., 2024). Their proposed gated attention coding (GAC) introduces an encoding layer with a gated attention unit that blends spatiotemporal information into the encoded spikes. However, the temporal attention in GAC is still extracted from repeated inputs, which lack feedback from previous timesteps.

## 3 DIRECT CODING ANALYSIS

### 3.1 PRELIMINARY

#### 3.1.1 SPIKING NEURON

The Leaky Integrate-and-Fire (LIF) model is a commonly used spiking neuron model, which is discretely formulated as follows:

$$\mathbf{U}^t = \tau \mathbf{V}^{t-1} + \mathbf{I}^t, \tag{1}$$

$$\mathbf{S}^t = \Theta(\mathbf{U}^t - \mathbf{V}_{\text{th}}), \tag{2}$$

$$\mathbf{V}^t = (1 - \mathbf{S}^t) \cdot \mathbf{U}^t + \mathbf{V}_{\text{rst}} \mathbf{S}^t. \tag{3}$$

$\mathbf{U}$ and $\mathbf{V}$ are the membrane potentials before and after spike generation, respectively. $\mathbf{I}$ is input feature from the preceding layer and $\tau \in [0, 1]$ is the decaying factor. $\mathbf{S} \in \{0, 1\}$ is spike determined by Heaviside step function $\Theta(\cdot)$ and threshold $\mathbf{V}_{\text{th}}$. When a spike occurs, the membrane potential is reset to the rest potential $\mathbf{V}_{\text{rst}}$. The timestep $t$ denotes the discrete moment when the variables are updated. The sequence of spikes generated over timesteps is referred to as a spike train, and the associated 0, 1 values are called its pattern.

#### 3.1.2 ENCODED SPIKE TRAINS IN DIRECT CODING

Under a direct coding scheme, the forward pass starts with a frame-based input passing through a typical ANN layer. The generated feature is accumulated in the spiking neuron at each timestep. The features obtained from the encoding layer can be expressed as follows:

$$\mathbf{I}^t = \mathbf{W}_{\text{enc}} \cdot \mathbf{X}^t, \tag{4}$$

where $\mathbf{W}_{\text{enc}}$ is the encoding layer weight and $\mathbf{X}^t$ is the input image at timestep $t$. In direct coding, the same inputs are repeatedly given to the encoding layer such that $\mathbf{X}^1$ through $\mathbf{X}^T$ all have the same value $\mathbf{X}$, with $T$ being the total simulated timestep. This also makes $\mathbf{I}^1$ to $\mathbf{I}^T$ to have the same value $\mathbf{I} = \mathbf{W}_{\text{enc}} \cdot \mathbf{X}$. At each timestep, $\mathbf{I}$ accumulates in $\mathbf{U}^t$ until it surpasses $\mathbf{V}_{\text{th}}$, and at that moment, a spike is fired. If a spiking neuron in the encoding layer generates its first spike at timestep $T_{\text{p}}$, $\mathbf{S}^{T_{\text{p}}}$ becomes 1 and $\mathbf{V}^{T_{\text{p}}}$ resets to $\mathbf{V}_{\text{rst}} (= 0)$. As $\mathbf{V}^{T_{\text{p}}}$ is 0 and the input is always the fixed value $\mathbf{I}$, the behavior of the spiking neuron from $T_{\text{p}} + 1$ to $2T_{\text{p}}$ exactly follows that from 1 to $T_{\text{p}}$, inducing the periodicity of output spike trains. The resulting period $T_{\text{p}}$ of the spike trains generated under a direct coding scheme is determined by the relationship between $\tau$, $\mathbf{V}_{\text{th}}$, and $\mathbf{I}$ (Qiu et al., 2024). The conditions that determine $T_{\text{p}}$ are formulated as follows:

$$\begin{cases} T_{\text{p}} = 1, & \text{if} \quad \mathbf{I} > \mathbf{V}_{\text{th}}, \\ T_{\text{p}} = k, & \text{if} \quad \frac{1-\tau}{1-\tau^k} \mathbf{V}_{\text{th}} < \mathbf{I} < \frac{1-\tau}{1-\tau^{k-1}} \mathbf{V}_{\text{th}}, \\ T_{\text{p}} = \infty & \text{if} \quad \mathbf{I} < (1 - \tau) \mathbf{V}_{\text{th}}. \end{cases} \tag{5}$$

The full derivation of the periodicity of direct coding and the length of its period is in Appendix A.

### 3.2 REPRESENTATIONAL POWER OF DIRECT CODING

#### 3.2.1 DISTRIBUTION IMBALANCE OF SPIKE TRAINS

Using Eq. 5 to gain insights into the distribution of spike trains, we visualize the relationship between $\mathbf{I}$ and $T_{\text{p}}$ in Fig. 1 (a). The ranges where $T_{\text{p}}$ equals 1, $k$, or $\infty$ correspond to regions C, B, and A, respectively. The patterns of the spike trains are written above each range, with the case of $T = 4$ as an example. It can be seen that the range of most $\mathbf{I}$ values corresponds to only a few patterns, such as $\langle 0000 \rangle$ (all-zero) or $\langle 1111 \rangle$ (all-one) patterns (regions A and C in Fig. 1 (a), respectively).

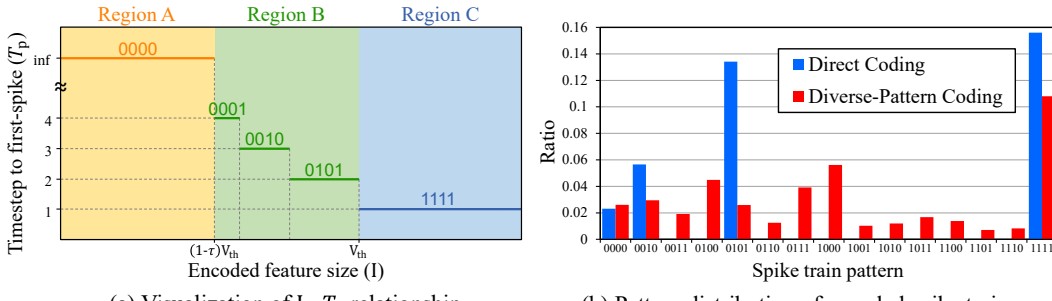

(a) Visualization of I - $T_p$ relationship      (b) Pattern distribution of encoded spike trains

Figure 1: (a) Relationship between the feature size ($\mathbf{I}$) and the length of spike train periods ($T_p$). (b) The output spike train distribution of the encoder layer under direct coding (DC) and our diverse-pattern coding (DPC). Ratio of the patterns relative to the total number of spike trains is represented, except for the all-zero pattern (0.63 and 0.57 for DC and DPC, respectively) for better visibility.

Based on this insight, we conduct experiments to observe the distribution of spike trains by probing the encoding layers of direct-coded SNN models. Fig. 1 (b) visualizes the pattern distribution of spike trains generated from the encoding layer of an SNN model (Spike-driven Transformer-2-512 (Yao et al., 2023) trained on CIFAR100 (Krizhevsky et al., 2009)) under a direct coding scheme and our proposed neural coding scheme, with details in Section 4. Experimental results on other SNN architectures and the exact number of spike trains for each pattern are in Appendix B, supporting the consistency of the observed trend across various models. We note that more than half of the spike trains correspond to the all-zero pattern (63%) due to the sparse nature of SNN. The all-one pattern, although not as dominant as the all-zero pattern, still occupies a large portion due to its broad range (region C in Fig. 1) (a). The portion of remaining periodic patterns that belong to region B declines dramatically as $T_p$ increases, and non-periodic patterns are never generated.

In summary, our findings show that the distribution of spike trains under a direct coding scheme is significantly imbalanced. Not only is the utilization of patterns limited by inherent periodicity, but the spike trains are also concentrated in a few specific patterns. Our analyses reveal that spike trains are not being generated *diversely* in the encoding layer due to the nature of direct coding.

### 3.2.2 SPIKE TRAIN DIVERSITY AND PERFORMANCE

Following the theoretical analysis, we also empirically investigate how the distribution of spike trains in the encoding layer affects the performance of SNN models. To this end, we explore the role of the decaying factor ($\tau$) and threshold ($V_{th}$), which determine the boundary of regions in Fig 1 (a). Adjusting these values allows us to control the distribution of spike trains. According to Fig. 1 (a), increasing $\tau$ causes the AB boundary to shift leftward, resulting in a narrowing of region A and a widening of region B, which alleviates distribution imbalance. Meanwhile, lowering $V_{th}$ causes both the AB and BC boundaries to shift leftward, narrowing regions A and B and widening region C. To quantitatively assess the differences in diversity between spike train distributions, the entropy of spike trains $\mathcal{Q}$ is measured as follows:

$$\mathcal{Q} = -\sum_{i=1}^{2^T} p\left(A^T = A_i^T\right) \log p\left(A^T = A_i^T\right), \tag{6}$$

where $A^T$ is the spike train up to timestep $T$, and $A_i^T$ is its specific pattern indexed by $i \in \{1, 2, \ldots, 2^T\}$. The validity of spike train entropy as a diversity metric is detailed in Appendix C.

We measure the classification accuracy of a Spike-driven Transformer-2-512 on the CIFAR100 dataset while varying $\tau$ and $V_{th}$ of the LIF neurons in the encoding layer to investigate the correlation between the diversity of spike trains and model performance under a direct coding scheme. Fig. 2 (a) and (b) show the empirical results on different values of $\tau$ and $V_{th}$, respectively. As visualized in (a-1) and (b-1), adjusting $\tau$ and $V_{th}$ enables altering the spike distribution. This change was quantified using spike train entropy, where (a-2) and (b-2) demonstrate that more imbalanced distributions correspond to lower entropy values. Both experiments lead to a shared conclusion regarding the relationship between entropy and accuracy: Models with higher spike train entropy tend to excel in model performance.

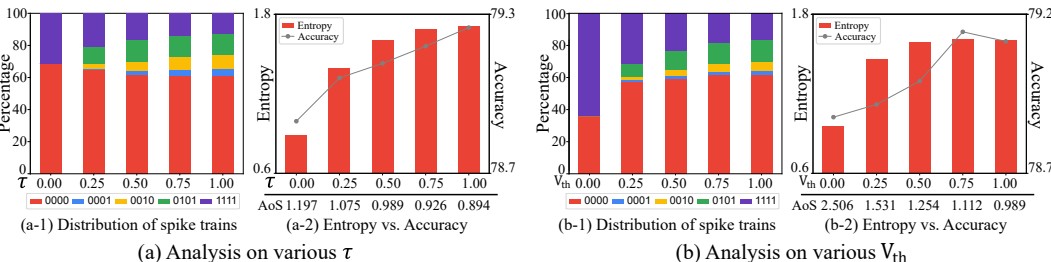

Figure 2: Spike train entropy and model performance with different $\tau$ and $\mathbf{V}_{th}$, along with pattern distributions of encoded spike trains. (a) Varying $\tau$. (b) Varying $\mathbf{V}_{th}$. The average number of spikes (AoS) is also reported.

It can also be deduced that the number of spikes or the periodicity of the encoding layer is not the primary issue in model performance. When comparing the average number of spikes (termed AoS) in Fig. 2, it is evident that, contrary to the common belief that an increase in the number of spikes generally benefits model performance (Sakemi et al., 2023), models with lower AoS demonstrated better results. This suggests that excessive imbalance among patterns has a more dominant negative impact on accuracy. Additionally, since all experiments are conducted under direct coding, periodicity remains inherently present, demonstrating that spike train entropy can interpret performance variations that cannot be accounted for by periodicity alone. Our analyses elucidate the phenomenon of spike train imbalance and highlight the importance of pattern diversity through the theoretically-derived repetitiveness of direct coding with empirical observations.

## 4 DIVERSE-PATTERN CODING

In the previous section, we showed that repeated inputs in direct coding lead to an imbalanced distribution of spike trains, hindering performance improvement. As a simple yet effective solution to enhance the pattern diversity of spike trains, we propose augmenting direct coding by explicitly injecting temporal information. Unlike direct coding, which treats inputs as temporally uniform, the proposed diverse-pattern coding (DPC) processes distinct information at each timestep, enabling a more diverse representation across the temporal dimension. As illustrated in Fig. 3, our DPC framework consists of two components: Temporal embedding (TE) and temporal feedback (TF) layer. Each injects temporal information into the inputs and encoded features, respectively. DPC can be seamlessly integrated into various models that previously employed direct coding, including convolutional neural networks (CNNs) and Transformers. In this section, we present the DPC framework and illustrate how it diversifies encoded spike trains with temporal dynamics.

### 4.1 TEMPORAL EMBEDDING

Temporal embedding (TE) is a learnable embedding tailored for SNNs designed to vary input data across different timesteps. TE generally enhances spike pattern diversity by *adding a learnable, time-dependent variation* to the input, so that the encoder observes a slightly different tensor at every timestep while the spatial content of the original frame is preserved. Based on Eq. 4, the dynamics of TE can be formulated as follows:

$$\mathbf{I}_{emb}^t = \mathbf{W}_{enc}\widetilde{\mathbf{X}}^t, \quad \text{where} \quad \widetilde{\mathbf{X}}^t = \mathbf{X}^t + \mathbf{E}^t. \tag{7}$$

Static frame $\mathbf{X} \in \mathbb{R}^{C \times H \times W}$ is replicated across $T$ timestep and perturbed by a learnable, time-dependent embedding $\mathbf{E} \in \mathbb{R}^{T \times C \times H \times W}$, resulting in an embedded input $\widetilde{\mathbf{X}}^t$. Our implementation resembles the learnable positional embeddings in video Transformers, where such embeddings serve as indicators of chronological order for representations from different frames (Zhang et al., 2023). Similarly, TE in DPC injects temporal variation to differentiate inputs across timesteps and avoid repetition. Inspired by the positional encoding designs from vision and video transformers (Hassani et al., 2021; Yuan et al., 2021; Zhang et al., 2023), we initialize TE with 3D sinusoidal positional encoding, the 3D generalization of the 2D sinusoidal scheme of (Wang & Liu, 2019). This 3D sinusoidal initialization encourages coherent spatio-temporal frequencies from the very first iteration, accelerating optimization without adding inference overhead. Let $\pi_0 = c$, $\pi_1 = h$, $\pi_2 = w$ and

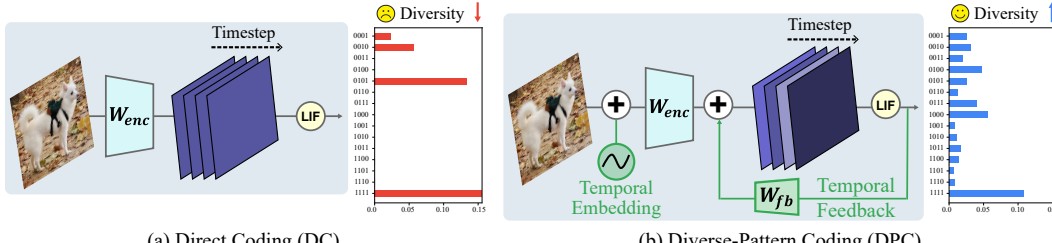

(a) Direct Coding (DC)        (b) Diverse-Pattern Coding (DPC)

Figure 3: Overview of (a) direct coding (DC) scheme and (b) diverse-pattern coding (DPC) scheme. Temporal embedding and temporal feedback layer of DPC provide time-varying information to the encoded spike trains, resulting in a more diverse distribution of spike train patterns.

define $d(t) = \lfloor 3t/T \rfloor$. For all indices $t, c, h, w$ TE is initialized as follows:

$$\mathbf{E}_{(0)}^{t,c,h,w} = \begin{cases} \sin\!\big(\omega_t\,\pi_{d(t)}\big), & t \text{ even}, \\ \cos\!\big(\omega_t\,\pi_{d(t)}\big), & t \text{ odd}, \end{cases} \qquad \omega_t = 10000^{-6\left\lfloor (t \bmod T/3)/2 \right\rfloor / T}, \tag{8}$$

where $\pi_{d(t)} \in \{c, h, w\}$ picks the coordinate according to $d(t) = \lfloor 3t/T \rfloor$. All elements are subsequently updated by direct training with surrogate gradients (Wu et al., 2018), allowing the optimizer to refine the temporal cues in an SNN-optimized, data-driven fashion. TE introduces effective temporal variation at the input stage, mitigating the repetition inherent in traditional direct coding.

## 4.2 TEMPORAL FEEDBACK LAYER

While TE reduces input repetition through learnable time-dependent embeddings, temporal dynamics can be further enriched by leveraging information from previous timesteps. To this end, we propose the Temporal feedback (TF) layer, an encoding layer with a recurrent feedback connection that targets enhancing the encoded features. Feedback is generated by projecting earlier spike output through feedback kernels and injecting it into the current features. Combining Eq. 1 with Eq. 7 and adding a feedback connection, the dynamics of the TF layer can be formulated as follows:

$$\mathbf{U}^t = \tau\mathbf{V}^{t-1} + \mathbf{I}_{\text{emb}}^t + \mathbf{W}_{\text{fb}} \cdot \mathbf{S}^{t-1}, \tag{9}$$

where $\mathbf{W}_{\text{fb}}$ represents feedback weights, configured identically to $\mathbf{W}_{\text{enc}}$ and optimized end-to-end along with other model parameters. At each timestep, the embedded input feature and the feedback from the preceding timestep apply temporal variation to the LIF neuron. TF layer enables interaction between subsequent timesteps, further enhancing the temporal information encoded in spike trains.

Although recurrent connections have previously been applied to SNNs (Shen et al., 2024; Zhang & Zhou, 2022), they have not been studied as a means of diversifying spike trains in neural coding. GAC (Qiu et al., 2024) utilizes temporal attention, but it works by squeezing repeated inputs on the temporal dimension and does not leverage information from past timesteps. DPC is the first to enhance temporal dynamics by incorporating information from previous timesteps directly within the neural coding stage. The effectiveness of DPC is shown in Fig. 1 (b), where the distribution imbalance of spike trains is successfully alleviated. The detailed algorithm is provided in Appendix E.

## 5 EXPERIMENTS

We evaluate the effectiveness of DPC across a diverse set of datasets and model architectures. DPC is applied in a plug-and-play manner, replacing direct coding as the encoding method, and we compare its performance against this baseline. Our experiments cover visual, time-series, and natural language understanding tasks to comprehensively assess the generality of DPC. In addition, we provide a detailed analysis of DPC's efficiency in terms of energy and memory consumption.

## 5.1 EXPERIMENTAL SETUP

We adopt direct training to train deep SNN models from scratch using spatio-temporal backpropagation (STBP) (Wu et al., 2018). To demonstrate the generalizability and effectiveness of our DPC, we

Table 1: Comparison of DPC with other encoding schemes on CIFAR datasets. SDT is a shortened term for Spike-driven Transformer and HST for Hierarchical Spiking Transformer used in QKFormer. Results are reported as classification accuracy (%).

| Architecture | Encoding | CIFAR10 | | | CIFAR100 | | |
|---|---|---|---|---|---|---|---|
| | | $T$=2 | $T$=4 | $T$=6 | $T$=2 | $T$=4 | $T$=6 |
| MS-ResNet-18 | DC | - | - | 94.58 | - | - | 76.80 |
| | GAC | 96.18 | 96.24 | 96.46 | 78.92 | 79.83 | 80.45 |
| | **DPC** | **96.39** | **96.79** | **96.81** | **80.23** | **80.82** | **81.04** |
| SDT-2-512 | DC | - | 95.6 | - | - | 78.4 | - |
| | **DPC** | **95.24** | **95.85** | **96.05** | **78.24** | **79.94** | **80.43** |
| HST-4-384 | DC | - | 96.18 | - | - | 81.15 | - |
| | **DPC** | **95.94** | **96.31** | **96.59** | **80.37** | **81.42** | **81.79** |

Table 2: Comparison of DPC with other encoding schemes on the ImageNet dataset. Encodings marked with [†] indicate results reproduced using the official code. The timestep is set to 4.

Table 3: Comparison of DPC with other encoding schemes on the CIFAR10-DVS dataset. The timestep is set to 10 and 16 for MS-ResNet and QKFormer, respectively.

| Architecture | Encoding | Acc. (%) |
|---|---|---|
| MS-ResNet-34 | DC[†] | 65.78 |
| | GAC[†] | 66.71 |
| | **DPC** | **69.04** |
| HST-10-384 | DC | 78.80 |
| | **DPC** | **79.22** |

| Architecture | Encoding | Acc. (%) |
|---|---|---|
| MS-ResNet-18 | DC | 78.2 |
| | **DPC** | **79.4** |
| HST-2-256 | DC | 84.0 |
| | **DPC** | **85.0** |

conduct experiments on diverse datasets, including static image datasets (CIFAR10/100 (Krizhevsky et al., 2009), ImageNet (Deng et al., 2009)), a neuromorphic dataset (CIFAR10-DVS (Li et al., 2017a)), time-series forecasting datasets (Metr-la (Li et al., 2017b), Electricity (Lai et al., 2018)), and natural language understanding datasets (MR (Pang & Lee, 2005), Subj (Pang & Lee, 2004), SST-5, SST-2 (Socher et al., 2013)). To further assess DPC's broad applicability, we evaluate it across multiple architectures, including CNN (MS-ResNet (Hu et al., 2024)) and Transformer variants (Spikformer (Zhou et al., 2023), Spike-driven Transformer (Yao et al., 2023), QKFormer (Zhou et al., 2024), iSpikformer (Lv et al., 2024)). More details on training our SNN models, including hyperparameters, architecture configuration, and data augmentation, can also be found in Appendix F.

## 5.2 BENCHMARK RESULTS

We demonstrate the plug-and-play ability of DPC to enhance spike-train diversity and improve SNN performance across a wide range of architectures and modalities. Compared to prior SNN encoding approach (Qiu et al., 2024), our experiments cover a broader set of datasets and model configurations, establishing DPC as a effective replacement for direct coding. As shown here and in Appendix G.2, we also compare DPC with several temporally variant encoding techniques, including GAC and other temporal augmentation baselines, and observe more stable and robust improvements. These results highlight not only the practical utility of DPC but also its theoretical insight by explicitly addressing the diversity bottleneck in SNNs, further underscoring the significance of our contribution. Further analyses, including ablations on individual DPC components and analysis of temporal information in the encoded spike trains, are presented in Appendices G and J.

**Image classification** We first evaluate DPC on CIFAR-10 and CIFAR-100. As shown in Table 1, DPC outperforms direct coding on both ResNet and Transformer architectures, even with fewer timesteps, highlighting its efficiency. In particular, DPC surpasses both the baseline MS-ResNet and GAC—the previous state-of-the-art neural coding method—with our timestep-4 model outperforming GAC's timestep-6 results on both benchmarks.

Table 4: Comparison of the proposed DPC with other encoding schemes on time-series benchmarks with four different prediction lengths (horizons). "Avg." denotes the mean across all 8 dataset–horizon pairs, and "Avg. Rank↓" indicates the average rank (lower is better) across these pairs. ↑(↓) indicates that higher(lower) is better. "*" denotes non-convergent cases.

| Encoding | Metric | Metr-la | | | | Electricity | | | | Avg. | Avg. Rank↓ |
|---|---|---|---|---|---|---|---|---|---|---|---|
| | | 6 | 24 | 48 | 96 | 6 | 24 | 48 | 96 | | |
| Repetition (DC) | $R^2\uparrow$ | .692 | .548 | .238 | .021* | .962 | .953 | .849 | .710* | .622 | 4.0 |
| | RSE↓ | .573 | .708 | .847 | 1.04* | .289 | .557 | .705 | 1.03* | .719 | 4.0 |
| Delta | $R^2\uparrow$ | .804 | .601 | .434 | .272 | .972 | .969 | .960 | .944 | .744 | 2.8 |
| | RSE↓ | .496 | .666 | .759 | .910 | .274 | .302 | .391 | .455 | .532 | 2.6 |
| Convolutional | $R^2\uparrow$ | .817 | .618 | **.440** | **.279** | .977 | .974 | .972 | .963 | .755 | 1.8 |
| | RSE↓ | .475 | .668 | **.752** | **.905** | .263 | .284 | .338 | .348 | .504 | 1.9 |
| **DPC** | $R^2\uparrow$ | **.847** | **.620** | .413 | .247 | **.991** | **.988** | **.984** | **.978** | **.759** | **1.5** |
| | RSE↓ | **.413** | **.653** | .808 | .915 | **.172** | **.195** | **.229** | **.264** | **.456** | **1.5** |

Table 5: Comparison of DPC with other encoding schemes on natural language understanding benchmarks. The timestep is fixed at 8, and results are reported as classification accuracy (%).

| Encoding | MR | SST-2 | Subj | SST-5 |
|---|---|---|---|---|
| DC | 76.27 | 81.19 | 90.65 | 42.08 |
| **DPC** | **77.20** | **82.22** | **92.50** | **43.26** |

We next validate DPC on ImageNet using MS-ResNet-34 and HST-10-384 models with a timestep of 4. As reported in Table 2, DPC consistently outperforms direct coding and GAC across both the ResNet and Transformer families. These improvements are achieved through a simple substitution of the encoder, underscoring the model-agnostic nature and broad applicability of DPC.

**Neuromorphic data classification** To validate the effectiveness of DPC beyond static images, we conduct experiments on CIFAR10-DVS, a standard neuromorphic benchmark, comparing against direct coding using MS-ResNet-18 and QKFormer. As shown in Table 3, DPC consistently outperforms direct coding, underscoring DPC's robustness across modalities. We further apply DPC to PSN (Fang et al., 2023), another high-performing SNN model. The reproduced PSN achieved 81.3% accuracy, while the DPC-augmented version achieved 82.3%. We emphasize that DPC is orthogonal to architectural innovation and can be complementarily integrated into powerful SNN models to potentially boost their performance even further.

**Time series forecasting** Time-series forecasting requires accurate temporal dependency modeling, making it a strong benchmark for evaluating SNNs' temporal processing capacity. Based on the iSpikformer architecture proposed by (Lv et al., 2024), we conduct experiments on two standard benchmarks: Metr-la and Electricity. We compare DPC with other encoding methods: repetition (equivalent to direct coding), delta and convolutional encoders, the latter two introduced in (Lv et al., 2024) to better capture intrinsic temporal structures. As shown in Table 4, DPC achieves the highest average $R^2$ and lowest average RSE across multiple prediction horizons, demonstrating superior temporal encoding capabilities. Notably, DPC significantly outperforms direct coding across all dataset–horizon combinations, highlighting the importance of spike pattern diversity and the strong potential of DPC for complex temporal tasks.

**Natural language understanding** To evaluate DPC in the language domain, we conduct experiments on standard natural language understanding benchmarks. Following the implementation of SpikeBERT (Lv et al., 2023), we train a Spikformer model with T = 8 using our DPC on four text classification datasets: MR, SST-2, Subj, and SST-5. All models are trained from scratch using direct training with surrogate gradients, and the results are summarized in Table 5. Across all datasets, DPC consistently outperforms direct coding. These results show that DPC generalizes beyond vision and time-series tasks, making it a versatile component for modeling diverse temporal domains.

Table 6: A comparative analysis of parameter sizes across two SDT architectures with DC and DPC schemes. The parameter counts for both schemes are reported in millions.

| Architecture | $T$ | DC | DPC |
|---|---|---|---|
| SDT-2-256 | 2 | 2.6 | 2.607 (+0.24%) |
| | 4 | 2.6 | 2.613 (+0.47%) |
| | 6 | 2.6 | 2.619 (+0.71%) |
| SDT-2-512 | 2 | 10.318 | 10.325 (+0.06%) |
| | 4 | 10.318 | 10.331 (+0.12%) |
| | 6 | 10.318 | 10.337 (+0.18%) |

Table 7: A comparative analysis of computational energy consumption of MS-ResNet-18 with DC and DPC schemes. The timestep is set to 6 and energy is reported in millijoules (mJ).

| Layer | | Op. type | DC | DPC |
|---|---|---|---|---|
| Encoding layer | $\mathbf{W}_{enc}$ | MAC | 0.049 | 0.049 |
| | TE | AC | - | $\sim$0 |
| | TF | AC | - | 0.053 |
| Deeper layers | | AC | 1.407 | 1.430 |
| Total | | - | 1.456 | 1.532 |

## 5.3 EFFICIENCY ANALYSIS

As DPC can be adopted by simply replacing the encoding layer, we consequently evaluate its efficiency in terms of parameter size, energy consumption, and latency.

Table 6 summarizes parameter counts for SNNs trained on CIFAR100 at timesteps 2, 4, and 6. Due to their structure design, TE parameters scale with timestep, and those of TF scale with the size of encoded features. SDT-2-256 and SDT-2-512 incur the same overhead for a fixed timestep (e.g., +0.007M at $T$=2), so the relative cost shrinks for larger models. Increasing $T$ raises overhead linearly. For comparison with other schemes, we also measure the model size with GAC. On MS-ResNet-18, DPC adds only +0.05M parameters at $T$=6 (0.44%) versus +0.13M for GAC (1.04%), highlighting DPC's memory efficiency.

We also quantify and report the energy overhead introduced by DPC. We perform the analysis on MS-ResNet-18 with CIFAR-100 at $T$=6, a representative moderately deep SNN architecture. Details on the calculation are provided in Appendix H, and the measured results are summarized in Table 7. TE incurs a negligible energy increase, and TF consumes energy comparable to the baseline DC encoder. Because the encoder accounts for only a small fraction of total model energy, the overall increase in end-to-end energy is approximately 5%. Moreover, since DPC modifies only the encoding stage, its relative contribution diminishes further in deeper networks. In summary, DPC yields consistent accuracy gains while adding only a marginal energy cost by virtue of TE and TF using only AC operations and introducing no new MACs, preserving the fundamental low-power compute pattern of SNNs.

For latency, we measure wall-clock time for both training and inference and report the results in Tables A12 and A13 of Appendix I. Depending on the architecture and timestep, DPC introduces only marginal slowdowns relative to DC. Training time increases by approximately 0.7%–7%, and inference time by about 2%–4%.

## 6 CONCLUSION

In this paper, we demonstrate that the widely used direct coding scheme for SNNs shows an imbalanced distribution of spike train patterns due to repeated input. We verify that this lack of diversity contributes to performance degradation through experiments examining the relationship between spike train entropy and performance. To improve the diversity of spike trains, we propose diverse-pattern coding (DPC), a novel neural coding scheme that integrates two key components: Temporal embedding (TE) and temporal feedback (TF) layer. Using this simple design, DPC injects temporal information into the original features without compromising their content, thereby effectively diversifying spike-train patterns. Extensive experiments on vision, neuromorphic, time-series, and language benchmarks confirm that applying DPC consistently improves performance across multiple architectural baselines while adding only marginal parameter and energy overhead. We believe our investigation will provide valuable insights into designing effective coding schemes for high-performing and efficient SNNs. Future research will focus on further improving energy efficiency to develop even more effective and lightweight neural coding.

## 7 REPRODUCIBILITY STATEMENT

We outline here the efforts made to ensure the reproducibility of our work. Details of the experimental setup are provided in Section 5.1 and Appendix F. All reported results are averaged over three random seeds, and the standard deviations for the main results are presented in Appendix D. A complete proof of the claims regarding the representational capacity limits of direct coding is included in Section 3 and Appendix A. Furthermore, a clear justification for using spike train entropy as a diversity metric is provided in Appendix C.

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

# A  PERIODICITY IN DIRECT CODING

We provide a detailed derivation showing that the encoded spike trains generated under the direct coding scheme are periodic. If a spiking neuron in the encoding layer generates its first spike at timestep $T_p$, i.e., no spike is generated for $T_p - 1$ timesteps, the membrane potentials until timestep $T - 1$ satisfies:

$$\mathbf{V}^t = \tau \mathbf{V}^{t-1} + \mathbf{I}^t \quad \text{for} \quad t \leq T_p - 1. \tag{A1}$$

Letting $t = T_p - 1$ and unfolding the recursive part for $\mathbf{V}$, we get:

$$\mathbf{V}^{T_p-1} = \tau^{T_p-1}\mathbf{V}^0 + \sum_{t=1}^{T_p-1} \tau^{T_p-t-1}\mathbf{I}^t, \tag{A2}$$

where $\mathbf{V}^0$ is the initial membrane potential, commonly set to 0. Because $\mathbf{I}^t = \mathbf{I}$ in direct coding as previously mentioned, $\mathbf{V}^{T_p-1}$ can be formulated as:

$$\mathbf{V}^{T_p-1} = \sum_{t=1}^{T_p-1} \tau^{T_p-t-1}\mathbf{I} = \frac{1 - \tau^{T_p-1}}{1 - \tau}\mathbf{I}. \tag{A3}$$

At timestep $T_p$, when a spike occurs, $\mathbf{V}^{T_p}$ resets to 0, and since the magnitude of the input remains constant, the dynamics described in Eq. A3 repeat periodically with a cycle of $T_p$.

Next, we derive the conditions that determine $T_p$. When an initial spike fires at the first timestep, $\mathbf{U}^1 > \mathbf{V}_{th}$, which is $\mathbf{I} > \mathbf{V}_{th}$. When an initial spike fires at timestep $k$, the membrane potential satisfies the range:

$$\mathbf{V}^{k-1} < \mathbf{V}_{th} < \mathbf{U}^k. \tag{A4}$$

Using Eq. A3 the condition becomes:

$$\frac{1 - \tau^{k-1}}{1 - \tau}\mathbf{I} < \mathbf{V}_{th} < \frac{1 - \tau^k}{1 - \tau}\mathbf{I}, \tag{A5}$$

and $\mathbf{I}$ lies in the range of:

$$\frac{1 - \tau}{1 - \tau^k}\boldsymbol{V}_{th} < \mathbf{I} < \frac{1 - \tau}{1 - \tau^{k-1}}\boldsymbol{V}_{th}. \tag{A6}$$

When no spikes are fired until the final timestep, the situation can be described as follows:

$$\lim_{T_p \to \infty} \mathbf{V}^{T_p-1} < \mathbf{V}_{th}. \tag{A7}$$

Simplifying the left-hand side yields the following expression:

$$\lim_{T_p \to \infty} \mathbf{V}^{T_p-1} = \lim_{T_p \to \infty} \frac{1 - \tau^{T_p-1}}{1 - \tau}\mathbf{I}$$
$$= \frac{\mathbf{I}}{1 - \tau}. \tag{A8}$$

Therefore, the condition for $\mathbf{I}$ that does not generate any spike comes down to:

$$\mathbf{I} < (1 - \tau)\,\boldsymbol{V}_{th}. \tag{A9}$$

# B  DISTRIBUTION OF ENCODED SPIKE TRAINS

## B.1  DISTRIBUTION FROM CNNS

We conduct experiments to observe the distribution of spike trains of the encoding layers on MS-ResNet. After training MS-ResNet-18 on the CIFAR100 dataset with direct coding and diverse-pattern coding (DPC), we plot the spike pattern distribution of the encoding layer in Fig. A1. Spike patterns generated from the encoding layer show similar consistency with those of Spike-driven Transformers. Distribution from direct training shows a significant imbalance, but DPC mitigates this lack of diversity to a large extent. These results suggest our DPC's effectiveness in improving temporal dynamics in neural coding, which applies to various architectures.

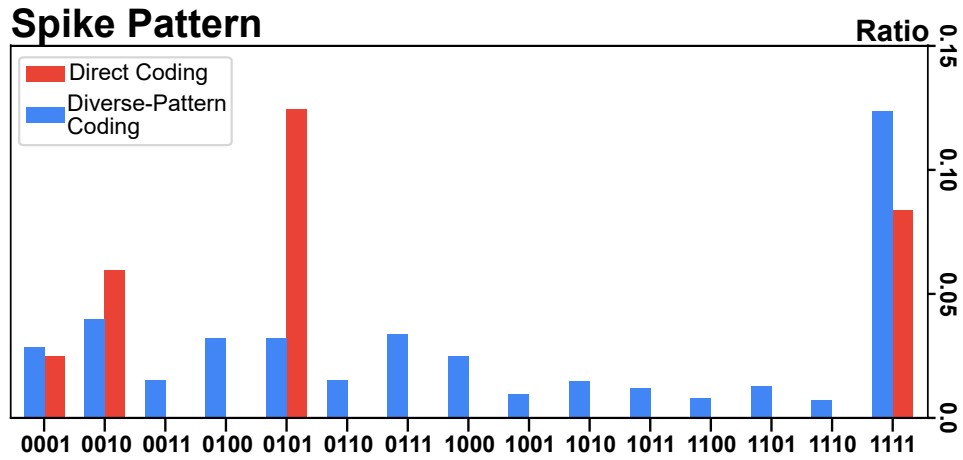

Figure A1: The output spike pattern distribution of the encoder layer of MS-ResNet-18, $T = 4$.

### B.2 SPIKE TRAIN COUNTS

We also report the actual number of appearances of each spike pattern. Tables A1, A2, and A3 lists the number of spikes generated from the encoding layer under direct coding and DPC scheme when $T$ equals 2, 4, and 6, respectively. The spike train entropy $\mathcal{Q}$ is also calculated to quantify the diversity of spike trains. The experiments were conducted with Spike-driven Transformer-2-512 trained on CIFAR100. The total sum is calculated as N * H * W * C, where N is the number of samples and H, W, and C are the height, width, and channel size of the encoded spike features, respectively. In our experiments, N=10000 (validation set size of CIFAR100), H=W=32, and C=64. It can be observed that for all timesteps, DPC generates significantly more diverse patterns compared to direct coding.

## C ANALYSIS ON SPIKE TRAIN ENTROPY

Following the analysis from the previous study (Qiu et al., 2024), the probability of a spike occurring at timestep $t$, given the spike train up to $t - 1$, can be calculated as follows:

$$p^t(a) = p\left(a^t = a \mid A^{t-1} = A_i^{t-1}\right), \tag{A10}$$

where $a^t \in [0, 1]$ is the spike value at $t$, $A^{t-1} = [a^1, a^2, \ldots, a^{t-1}]$ is the spike train up to timestep $t - 1$, and $A_i^{t-1}$ is an instance of $A^{t-1}$ indexed by $i \in \{1, 2, \ldots, 2^{t-1}\}$. Note that $A_1^0 = \emptyset$ and $p(A^0 = A_1^0) = 1$. The entropy value of $p^t(a)$ distribution, which we denote the one-step entropy $H(t)$, can be expressed as follows:

$$H(t) = -\sum_{a=0,1} p^t(a) \log p^t(a). \tag{A11}$$

A higher $H(t)$ value indicates that more information has been encoded at step $t$. By performing a weighted summation of the one-step entropy at each timestep with respect to the frequency of the spike train generated until the previous timestep, the resulting measure can serve as a proxy for the total quantity of information. This can be formulated as follows:

$$\mathcal{Q}_{\text{info}} = \sum_{t=1}^{T} \sum_{i=1}^{2^{t-1}} p\left(A^{t-1} = A_i^{t-1}\right) H(t). \tag{A12}$$

Eq A12 is equivalent to the entropy of the complete spike trains:

$$\mathcal{Q}_{\text{div}} = -\sum_{i=1}^{2^T} p\left(A^T = A_i^T\right) \log p\left(A^T = A_i^T\right). \tag{A13}$$

Table A1: Spike train counts and spike train entropy when $T = 2$.

| Patt. | Direct Coding | | Diverse-pattern Coding | |
|---|---|---|---|---|
| | Count | Ratio | Count | Ratio |
| $\langle 00 \rangle$ | 483,891,801 | 0.738 | 410,279,828 | 0.626 |
| $\langle 01 \rangle$ | 85,673,169 | 0.131 | 71,538,437 | 0.109 |
| $\langle 10 \rangle$ | 0 | 0 | 55,150,351 | 0.084 |
| $\langle 11 \rangle$ | 85,795,030 | 0.131 | 118,391,384 | 0.181 |
| Sum | 655,360,000 | 1 | 655,360,000 | 1 |
| $\mathcal{Q}$ | - | 1.091 | - | **1.518** |

Table A2: Spike train counts and spike train entropy when $T = 4$.

| Patt. | Direct Coding | | Diverse-pattern Coding | |
|---|---|---|---|---|
| | Count | Ratio | Count | Ratio |
| $\langle 0000 \rangle$ | 413,090,768 | 0.630 | 373,974,960 | 0.571 |
| $\langle 0001 \rangle$ | 15,131,991 | 0.023 | 17,100,070 | 0.026 |
| $\langle 0010 \rangle$ | 37,040,305 | 0.057 | 19,357,752 | 0.030 |
| $\langle 0011 \rangle$ | 0 | 0 | 12,524,515 | 0.019 |
| $\langle 0100 \rangle$ | 0 | 0 | 29,443,087 | 0.045 |
| $\langle 0101 \rangle$ | 87,890,854 | 0.134 | 16,954,674 | 0.026 |
| $\langle 0110 \rangle$ | 0 | 0 | 8,169,607 | 0.012 |
| $\langle 0111 \rangle$ | 0 | 0 | 25,660,264 | 0.039 |
| $\langle 1000 \rangle$ | 0 | 0 | 36,779,298 | 0.056 |
| $\langle 1001 \rangle$ | 0 | 0 | 6,687,881 | 0.01 |
| $\langle 1010 \rangle$ | 0 | 0 | 7,847,649 | 0.012 |
| $\langle 1011 \rangle$ | 0 | 0 | 10,972,251 | 0.017 |
| $\langle 1100 \rangle$ | 0 | 0 | 9,077,695 | 0.014 |
| $\langle 1101 \rangle$ | 0 | 0 | 4,663,172 | 0.007 |
| $\langle 1110 \rangle$ | 0 | 0 | 5,381,946 | 0.008 |
| $\langle 1111 \rangle$ | 102,206,082 | 0.156 | 70,765,179 | 0.108 |
| Sum | 655,360,000 | 1 | 655,360,000 | 1 |
| $\mathcal{Q}$ | - | 1.586 | - | **2.474** |

We demonstrate the equivalence through mathematical induction.

*Base case*: If $T = 1$, $\mathcal{Q}_{\text{info}} = \mathcal{Q}_{\text{div}} = H(1)$.

*Induction step*: Assume $\mathcal{Q}_{\text{info}} = \mathcal{Q}_{\text{div}}$ for $T = k$. Then, $\mathcal{Q}_{\text{info}}$ for $T = k + 1$ becomes:

$$
\begin{aligned}
\mathcal{Q}_{\text{info}} &= \sum_{t=1}^{k+1} \sum_{i=1}^{2^{t-1}} p\left(A^{t-1} = A_i^{t-1}\right) H(t) \\
&= \sum_{i=1}^{2^k} p\left(A^k = A_i^k\right) H(k+1) + \sum_{t=1}^{k} \sum_{i=1}^{2^{t-1}} p\left(A^{t-1} = A_i^{t-1}\right) H(t). \\
&= \sum_{i=1}^{2^k} p\left(A^k = A_i^k\right) H(k+1) - \sum_{i=1}^{2^k} p\left(A^k = A_i^k\right) \log p\left(A^k = A_i^k\right).
\end{aligned}
\tag{A14}
$$

Table A3: Spike train counts and spike train entropy when $T = 6$.

| Patt. | Direct Coding | | Diverse-pattern Coding | | Patt. | Direct Coding | | Diverse-pattern Coding | |
|---|---|---|---|---|---|---|---|---|---|
| | Count | Ratio | Count | Ratio | | Count | Ratio | Count | Ratio |
| $\langle 000000 \rangle$ | 430,515,704 | 0.657 | 326,637,665 | 0.498 | $\langle 100000 \rangle$ | 0 | 0 | 29,111,606 | 0.044 |
| $\langle 000001 \rangle$ | 4,161,034 | 0.006 | 7,791,219 | 0.012 | $\langle 100001 \rangle$ | 0 | 0 | 2,835,417 | 0.004 |
| $\langle 000010 \rangle$ | 8,614,496 | 0.013 | 8,001,636 | 0.012 | $\langle 100010 \rangle$ | 0 | 0 | 2,313,028 | 0.004 |
| $\langle 000011 \rangle$ | 0 | 0 | 5,536,172 | 0.008 | $\langle 100011 \rangle$ | 0 | 0 | 1,479,110 | 0.002 |
| $\langle 000100 \rangle$ | 18,325,143 | 0.028 | 10,927,392 | 0.017 | $\langle 100100 \rangle$ | 0 | 0 | 2,015,643 | 0.003 |
| $\langle 000101 \rangle$ | 0 | 0 | 4,615,266 | 0.007 | $\langle 100101 \rangle$ | 0 | 0 | 2,111,363 | 0.003 |
| $\langle 000110 \rangle$ | 0 | 0 | 2,600,847 | 0.004 | $\langle 100110 \rangle$ | 0 | 0 | 839,095 | 0.001 |
| $\langle 000111 \rangle$ | 0 | 0 | 5,985,575 | 0.009 | $\langle 100111 \rangle$ | 0 | 0 | 1,654,197 | 0.003 |
| $\langle 001000 \rangle$ | 0 | 0 | 13,006,876 | 0.020 | $\langle 101000 \rangle$ | 0 | 0 | 2,641,130 | 0.004 |
| $\langle 001001 \rangle$ | 39,774,954 | 0.061 | 3,479,252 | 0.005 | $\langle 101001 \rangle$ | 0 | 0 | 949,179 | 0.001 |
| $\langle 001010 \rangle$ | 0 | 0 | 6,698,685 | 0.01 | $\langle 101010 \rangle$ | 0 | 0 | 1,979,267 | 0.003 |
| $\langle 001011 \rangle$ | 0 | 0 | 3,018,633 | 0.005 | $\langle 101011 \rangle$ | 0 | 0 | 2,080,959 | 0.003 |
| $\langle 001100 \rangle$ | 0 | 0 | 3,549,591 | 0.005 | $\langle 101100 \rangle$ | 0 | 0 | 887,118 | 0.001 |
| $\langle 001101 \rangle$ | 0 | 0 | 3,560,558 | 0.005 | $\langle 101101 \rangle$ | 0 | 0 | 1,460,738 | 0.002 |
| $\langle 001110 \rangle$ | 0 | 0 | 2,940,594 | 0.004 | $\langle 101110 \rangle$ | 0 | 0 | 1,153,658 | 0.002 |
| $\langle 001111 \rangle$ | 0 | 0 | 8,278,173 | 0.013 | $\langle 101111 \rangle$ | 0 | 0 | 6,423,709 | 0.010 |
| $\langle 010000 \rangle$ | 0 | 0 | 17,357,517 | 0.026 | $\langle 110000 \rangle$ | 0 | 0 | 3,512,156 | 0.005 |
| $\langle 010001 \rangle$ | 0 | 0 | 2,762,294 | 0.004 | $\langle 110001 \rangle$ | 0 | 0 | 333,269 | 0.001 |
| $\langle 010010 \rangle$ | 0 | 0 | 4,694,922 | 0.007 | $\langle 110010 \rangle$ | 0 | 0 | 430,428 | 0.001 |
| $\langle 010011 \rangle$ | 0 | 0 | 1,683,660 | 0.003 | $\langle 110011 \rangle$ | 0 | 0 | 406,883 | 0.001 |
| $\langle 010100 \rangle$ | 0 | 0 | 5,460,468 | 0.008 | $\langle 110100 \rangle$ | 0 | 0 | 1,195,241 | 0.002 |
| $\langle 010101 \rangle$ | 77,195,618 | 0.118 | 5,289,294 | 0.008 | $\langle 110101 \rangle$ | 0 | 0 | 739,321 | 0.001 |
| $\langle 010110 \rangle$ | 0 | 0 | 3,098,299 | 0.005 | $\langle 110110 \rangle$ | 0 | 0 | 591,565 | 0.001 |
| $\langle 010111 \rangle$ | 0 | 0 | 4,260,721 | 0.007 | $\langle 110111 \rangle$ | 0 | 0 | 1,858,898 | 0.003 |
| $\langle 011000 \rangle$ | 0 | 0 | 3,835,034 | 0.006 | $\langle 111000 \rangle$ | 0 | 0 | 2,154,117 | 0.003 |
| $\langle 011001 \rangle$ | 0 | 0 | 1,150,057 | 0.002 | $\langle 111001 \rangle$ | 0 | 0 | 268,225 | 0.000 |
| $\langle 011010 \rangle$ | 0 | 0 | 3,028,255 | 0.005 | $\langle 111010 \rangle$ | 0 | 0 | 811,764 | 0.001 |
| $\langle 011011 \rangle$ | 0 | 0 | 3,079,647 | 0.005 | $\langle 111011 \rangle$ | 0 | 0 | 1,048,501 | 0.002 |
| $\langle 011100 \rangle$ | 0 | 0 | 2,597,938 | 0.004 | $\langle 111100 \rangle$ | 0 | 0 | 2,351,544 | 0.004 |
| $\langle 011101 \rangle$ | 0 | 0 | 3,233,555 | 0.005 | $\langle 111101 \rangle$ | 0 | 0 | 1,227,837 | 0.002 |
| $\langle 011110 \rangle$ | 0 | 0 | 3,649,535 | 0.006 | $\langle 111110 \rangle$ | 0 | 0 | 2,729,828 | 0.004 |
| $\langle 011111 \rangle$ | 0 | 0 | 30,527,987 | 0.047 | $\langle 111111 \rangle$ | 76,773,051 | 0.117 | 63,427,889 | 0.097 |
| | | | | | Sum | 655,360,000 | 1 | 655,360,000 | 1 |
| | | | | | $\mathcal{Q}$ | - | 1.642 | - | **3.462** |

The first term inside the summation can be rewritten as follows:

$$
\begin{aligned}
p\left(A^k = A_i^k\right) H(k+1) &= -\sum_{a=0,1} p\left(A^{k+1} = \left[A_i^k, a\right]\right) \log p\left(a^{k+1} = a \mid A^k = A_i^k\right) \\
&= -\sum_{a=0,1} p\left(A^{k+1} = \left[A_i^k, a\right]\right) \log p\left(A^{k+1} = \left[A_i^k, a\right]\right) \\
&\quad + \sum_{a=0,1} p\left(A^{k+1} = \left[A_i^k, a\right]\right) \log p\left(A^k = A_i^k\right) \\
&= -\sum_{a=0,1} p\left(A^{k+1} = \left[A_i^k, a\right]\right) \log p\left(A^{k+1} = \left[A_i^k, a\right]\right) \\
&\quad + p\left(A^k = A_i^k\right) \log p\left(A^k = A_i^k\right).
\end{aligned}
\tag{A15}
$$

Combining Eq. A14 and Eq. A15, we can get the final result as follows:

$$
\begin{aligned}
\mathcal{Q}_{\text{info}} &= -\sum_{i=1}^{2^{k+1}} p\left(A^{k+1} = A_i^{k+1}\right) \log p\left(A^{k+1} = A_i^{k+1}\right) \\
&= \mathcal{Q}_{\text{div}},
\end{aligned}
\tag{A16}
$$

showing that $\mathcal{Q}_{\text{info}} = \mathcal{Q}_{\text{div}}$ holds for $T = k + 1$. Therefore, we prove by mathematical induction that the proposition holds for all $T \geq 1$, implying that the spike train entropy, which quantifies the pattern diversity of spike trains, is equivalent to the cumulative information added at each timestep. In other words, more diverse spike trains encode more information, which is then transmitted to deeper layers, resulting in improved model performance.

Table A4: Mean and standard deviation results of DPC on image classification tasks over three runs with different seeds. SDT is a shortened term for Spike-driven Transformer and HST for Hierarchical Spiking Transformer. $T$ represents the simulated timesteps.

| Dataset | Architecture | Acc. (%) | | |
|---|---|---|---|---|
| | | $T$=2 | $T$=4 | $T$=6 |
| CIFAR10 | MS-ResNet-18 | 96.39±0.08 | 96.79±0.13 | 96.81±0.07 |
| | SDT-2-512 | 95.24±0.12 | 95.85±0.10 | 96.05±0.09 |
| | HST-4-384 | 95.94±0.06 | 96.31±0.07 | 96.59±0.03 |
| CIFAR100 | MS-ResNet-18 | 80.23±0.03 | 80.82±0.14 | 81.04±0.15 |
| | SDT-2-512 | 78.24±0.18 | 79.94±0.13 | 80.43±0.13 |
| | HST-4-384 | 80.37±0.15 | 81.42±0.12 | 81.79±0.10 |
| ImageNet | MS-ResNet-34 | - | 69.04±0.14 | - |
| | HST-10-384 | - | 79.22±0.11 | - |

Table A5: Mean and standard deviation results of DPC on neuromorphic data classification tasks over three runs with different seeds.

| Dataset | Architecture | Timestep | Acc. (%) |
|---|---|---|---|
| CIFAR10-DVS | MS-ResNet-18 | 10 | 79.40±0.12 |
| | HST-2-256 | 16 | 85.00±0.05 |

## D EXPERIMENT STATISTICS

We report the mean and standard deviation over three runs with different random seeds for all main experiments. Detailed statistics corresponding to Tables 1–5 in the main text are provided in Tables A4–A7 in the appendix.

## E DIVERSE-PATTERN CODING ALGORITHM

In this section, we provide the pseudo-code for the proposed DPC algorithm in Algorithm A1.

---

**Algorithm A1** Diverse-pattern coding (DPC) algorithm

---

**Input:** static frame data: $\mathbf{X} \in \mathbb{R}^{C \times H \times W}$; simulated timestep: $T$; neuron hyperparameters: (decaying factor $\tau$, firing threshold $V_{\text{th}}$, rest potential $V_{\text{rst}}$)
**Output:** spike train $\{S^t\}_{t=1}^{T}$
1: Initialize membrane $V^0 \leftarrow 0$, spike $S^0 \leftarrow 0$
2: Initialize temporal embedding $\mathbf{E}$ by Eq. equation 8
3: Initialize encoder weights $W_{\text{enc}}$, feedback weights $W_{\text{fb}}$
4: **for** $t \leftarrow 1$ **to** $T$ **do**
5: $\quad \widetilde{\mathbf{X}}^t \leftarrow \mathbf{X} + \mathbf{E}^t$ // Eq. equation 7
6: $\quad \mathbf{I}_{\text{emb}}^t \leftarrow W_{\text{enc}} \widetilde{\mathbf{X}}^t$
7: $\quad$ **if** $t = 1$ **then**
8: $\quad\quad U^t \leftarrow \mathbf{I}_{\text{emb}}^t$
9: $\quad$ **else**
10: $\quad\quad U^t \leftarrow \tau V^{t-1} + \mathbf{I}_{\text{emb}}^t + W_{\text{fb}} S^{t-1}$ // TF, Eq. equation 9
11: $\quad$ **end if**
12: $\quad S^t \leftarrow \Theta(U^t - V_{\text{th}})$ // spike generation
13: $\quad V^t \leftarrow (1 - S^t) U^t + S^t V_{\text{rst}}$ // membrane reset
14: **end for**

---

Table A6: Mean and standard deviation results of DPC on time series forecasting tasks over three runs with different seeds. The iSpikformer model is used with a timestep of 4.

| Metric | Metr-la | | | | Electricity | | | |
|---|---|---|---|---|---|---|---|---|
| | 6 | 24 | 48 | 96 | 6 | 24 | 48 | 96 |
| $R^2_\uparrow$ | .847±.003 | .620±.003 | .413±.009 | .247±.015 | .991±.001 | .988±.001 | .984±.003 | .978±.006 |
| $RSE_\downarrow$ | .413±.002 | .653±.003 | .808±.010 | .915±.009 | .172±.006 | .195±.007 | .229±.010 | .264±.008 |

Table A7: Mean and standard deviation results of DPC on natural language understanding tasks over three runs with different seeds. The Spikformer model is used with a timestep of 8.

| MR | SST-2 | Subj | SST-5 |
|---|---|---|---|
| 77.20±0.49 | 82.22±0.23 | 92.50±0.93 | 43.26±0.56 |

## F  IMPLEMENTATION DETAILS

We describe the implementation details for our experiments. Parameters not explicitly specified, such as those other than TE, are initialized following the default methods provided by standard libraries, typically drawn from normal distributions. Hyperparameters and training settings for MS-ResNet and Transformer models are in Table A8. We follow the implementation standards established in the previous papers (Hu et al., 2024; Lv et al., 2023; 2024; Qiu et al., 2024; Yao et al., 2023; Zhou et al., 2024). MS-ResNet-18, iSpikformer, Spikformer experiments are conducted on NVIDIA A40 GPUs, SDT and QKFormer experiments on NVIDIA V100 GPUs, and MS-ResNet-34 experiments on NVIDIA H100 GPUs.

### F.1  IMAGE CLASSIFICATION DETAILS

For direct training of MS-ResNet models on CIFAR and ImageNet datasets, we follow the surrogate gradient training convention from the original MS-ResNet paper (Hu et al., 2024). We use the data augmentation policies from the MS-ResNet implementation of GAC (Qiu et al., 2024), including CutMix and AutoAugment. For a fair comparison, we follow the architecture configuration from (Hu et al., 2024; Qiu et al., 2024), as in Table A9.

For experiments on Transformers, we adopt Spike-driven Transformer (Yao et al., 2023) and QK-Former (Zhou et al., 2024) for our baselines. We experiment with SDT-2-512, HST-4-384, and HST-10-384 models, in which the first and second numbers represent the number of encoder blocks and channels, respectively. For a fair comparison with the baseline model, we follow the implementation convention from the original papers, including surrogate gradient learning and architecture configuration.

### F.2  NEUROMORPHIC DATA CLASSIFICATION DETAILS

For QKFormer experiments on the CIFAR10-DVS dataset, we strictly follow the original implementation, including the simulation timestep of 16. For MS-ResNet-18, we adopt the same training configuration used in our CIFAR experiments, except for the simulation timestep, which is set to 10 following the settings in (Guo et al., 2022). For PSN experiment, we followed the same experimental setup as the original paper (Fang et al., 2023). Although we were unable to exactly replicate the original performance due to missing implementation details such as the train-test split ratio, we carefully reproduced the training pipeline to match the original configuration as closely as possible.

### F.3  TIME SERIES FORECASTING DETAILS

To evaluate the time-series forecasting capabilities of our proposed DPC, we conduct experiments on two widely used benchmarks:

Table A8: Hyperparameter and training settings for MS-ResNet and Transformer-based models. MS-ResNet-18 and SDT-2-512/HST-4-384 are for CIFAR; MS-ResNet-34 and HST-10-384 are for ImageNet.

|  | MS-ResNet-18 | MS-ResNet-34 | SDT-2-512 | HST-4-384 | HST-10-384 |
|---|---|---|---|---|---|
| $V_{\mathrm{th}}$ | 1 | 0.5 | 1 | 1 | 1 |
| $V_{\mathrm{rst}}$ | 0 | 0 | 0 | 0 | 0 |
| $\tau$ | 0.5 | 0.25 | 0.5 | 0.5 | 0.5 |
| $lr$ | 0.1 | 0.1 | 3e-4 | 1e-3 | 6e-4 |
| batch size | 128 | 256 | 64 | 64 | 64 |
| epoch | 250 | 300 | 300 | 500 | 200 |
| weight decay | 5e-5 | 1e-5 | 6e-2 | 6e-2 | 5e-2 |
| optimizer | SGD | SGD | AdamW | AdamW | AdamW |
| lr scheduler | CosineAnnealingLR | | | | |

Table A9: Architecture details of MS-ResNet models.

| Stage | MS-ResNet-18 | MS-ResNet-34 |
|---|---|---|
| Conv1 | 3x3, 64, stride=1 | 7x7, 64, stride=2 |
| Conv2 |  | $\begin{bmatrix} 3\text{x}3,\ 64 \\ 3\text{x}3,\ 64 \end{bmatrix} * 3$ |
| Conv3 | $\begin{bmatrix} 3\text{x}3,\ 128 \\ 3\text{x}3,\ 128 \end{bmatrix} * 3$ | $\begin{bmatrix} 3\text{x}3,\ 128 \\ 3\text{x}3,\ 128 \end{bmatrix} * 4$ |
| Conv4 | $\begin{bmatrix} 3\text{x}3,\ 256 \\ 3\text{x}3,\ 256 \end{bmatrix} * 3$ | $\begin{bmatrix} 3\text{x}3,\ 256 \\ 3\text{x}3,\ 256 \end{bmatrix} * 6$ |
| Conv5 | $\begin{bmatrix} 3\text{x}3,\ 512 \\ 3\text{x}3,\ 512 \end{bmatrix} * 2$ | $\begin{bmatrix} 3\text{x}3,\ 512 \\ 3\text{x}3,\ 512 \end{bmatrix} * 3$ |
| FC | AveragePool, FC | |

- **Metr-la**(Li et al., 2017b): Contains average traffic speed readings collected from sensors deployed across highways in Los Angeles County.
- **Electricity**(Lai et al., 2018): Consists of hourly electricity consumption data (in kWh).

We adopt iSpikformer (Lv et al., 2024), a spiking variant of iTransformer specifically designed to handle time-series data. We use two standard metrics for evaluation: coefficient of determination ($R^2$) and root relative squared error (RSE). Our implementation strictly follows the official settings and framework provided in (Lv et al., 2024), including dataset preprocessing, simulation timestep of 4, 2 encoding blocks, and a feature dimension of 512. These configurations are consistently applied across all experiments to ensure fair comparisons.

### F.4 NATURAL LANGUAGE UNDERSTANDING DETAILS

To evaluate the natural language understanding (NLU) capability of our proposed DPC, we conduct experiments on four widely used text classification benchmarks:

- **MR** (Pang & Lee, 2005): The Movie Review (MR) dataset contains movie-review sentences labeled for binary sentiment classification (positive or negative).
- **Subj** (Pang & Lee, 2004): This dataset consists of 10,000 sentences from movie reviews and plot summaries, labeled as subjective or objective for binary classification.
- **SST-5** (Socher et al., 2013): The Stanford Sentiment Treebank (SST-5) includes 11,855 movie-review sentences annotated with five sentiment categories: very negative, negative, neutral, positive, and very positive.
- **SST-2**: The binary version of SST-5, containing only positive and negative sentiment labels.

Table A10: The contribution of each component of DPC.

| Type | TE | TF layer | CIFAR100 | Tiny ImageNet | |
| | | | SDT-2-512 | MS-ResNet-18 | MS-ResNet-34 |
|---|---|---|---|---|---|
| (1) | | | 79.12% | 49.10% | 51.44% |
| (2) | ✓ | | 79.21% | 49.22% | 52.40% |
| (3) | | ✓ | 79.44% | 50.50% | 52.90% |
| (4) | ✓ | ✓ | **79.94%** | **51.04%** | **53.36%** |

Table A11: Performance of other time-variant encoding strategies.

| Method | Random spatial seg. | Directed spatial seg. | Random noise injection | DPC |
|---|---|---|---|---|
| Acc. | 95.73% | 95.91% | 96.10% | **96.81%** |

Our implementation strictly follows the official settings and framework described in (Lv et al., 2023), employing the Spikformer (Zhou et al., 2023) architecture with a simulation timestep of 8, encoder depth of 6, and feature dimension of 768. These configurations are applied consistently across all datasets to ensure fair and reproducible comparisons.

# G  ABLATION STUDIES

## G.1  COMPONENT ANALYSIS

We examine the impact of each component in the DPC scheme on two datasets. One of the datasets is CIFAR100, trained using SDT-2-512 with a timestep of 4, which was also used in the main results. The other is Tiny ImageNet, a subset of ImageNet comprising 100,000 images across 200 classes, resized to 64×64 colored images. We employ minimal data augmentation and training techniques and trained the models for 250 epochs with a timestep of 4. The results are reported in Table A10. Entropy measures show that compared to the baseline (1), which uses vanilla direct coding, those utilizing TE (2) and TF (3) each generate significantly more diverse spike trains. The model with both TE and TF (4) showed the highest performance improvement, emphasizing the combined effect of the two components.

## G.2  ALTERNATIVE TIME-VARIANT ENCODING STRATEGIES

In this section, we discuss other trials that also increase the pattern diversity of spike trains. We devised a method where images were segmented to ensure that spatially distinct inputs were provided at each timestep, compared to the temporal variation of DPC. We investigated two encoding strategies accordingly. In the first approach, we apply two augmentation policies, *RandomCrop* and *RandomHorizontalFlip*, to the input image at each timestep using different random seeds, allowing the model to randomly focus on different regions of the input over time. In the second approach, we reduce randomness and shift the bounding box of the CutMix augmentation along a fixed path, allowing the model to examine the entire region of the input over all timesteps. We refer to the first approach as *random spatial segmentation* and the second as *directed spatial segmentation*. We also designed a scheme in which random noise is injected at each timestep, assuming a scenario where variance is added across timesteps without any temporal information. To explore this idea, we conducted an experiment where Gaussian noise was added at each timestep for both training and inference, i.e., $\mathbf{I}^t = \mathbf{W}_{\text{enc}} \cdot \mathbf{X}^t + \epsilon^t$, where $\epsilon^t \sim N(0, s^2)$.

The experimental results are reported in Table A11. We use MS-ResNet-18 on the CIFAR10 dataset with a timestep of 6. While the first two strategies introduce spatial variations across time, their performances were inferior to our proposed DPC, which incorporates temporal embedding and feedback. It can be observed that the SNN encoder struggles to effectively process spatially dynamic inputs at each timestep. The third strategy also showed lower performance compared to DPC, highlighting the importance of temporal information. This approach also requires an additional hyperpa-

Table A12: Training latency per epoch (in seconds) for direct coding (DC), DPC, and GAC schemes across datasets, architectures, and timesteps. The values in parentheses indicate the relative increase in training time compared to the direct coding scheme.

| Dataset | Architecture | $T$ | DC | DPC | GAC |
|---|---|---|---|---|---|
| CIFAR100 | MS-ResNet-18 | 2 | 104.6 | 109.2 (+4.4%) | 122.6 (+17.2%) |
| | MS-ResNet-18 | 4 | 238.8 | 241.9 (+1.3%) | 259.8 (+8.8%) |
| | MS-ResNet-18 | 6 | 400.8 | 407.2 (+1.6%) | 429.3 (+7.1%) |
| | HST-4-384 | 2 | 68.4 | 68.9 (+0.7%) | - |
| | HST-4-384 | 4 | 103.1 | 108.2 (+4.9%) | - |
| | HST-4-384 | 6 | 142.9 | 148.2 (+3.7%) | - |
| ImageNet | HST-10-384 | 4 | 1184.4 | 1267.6 (+7.0%) | - |

Table A13: Inference latency of validation set (in seconds) for DC and DPC schemes across timesteps. The values in parentheses indicate the relative increase in inference time compared to the direct coding scheme.

| Dataset | Architecture | $T$ | DC | DPC |
|---|---|---|---|---|
| CIFAR100 | HST-4-384 | 2 | 4.3 | 4.4 (+2.3%) |
| | | 4 | 7.6 | 7.9 (+3.9%) |
| | | 6 | 11.1 | 11.5 (+3.6%) |

rameter (noise intensity $s$), which involves tuning, and since the optimal value can vary depending on the dataset or model, it must be manually adjusted in each case ($s = 1$ in this experiment). In contrast, DPC introduces learnable temporal variation through TE and TF, enabling more controlled spike train diversity that better integrates with the SNN architecture. By presenting the entire input throughout all timesteps and incorporating temporal variation to prevent repetition, our DPC best fits the characteristics of SNNs among diverse strategies.

### G.3 ENTROPY REGULARIZATION IN THE OBJECTIVE FUNCTION

To further improve the diversity of spike trains, we experimented with incorporating an entropy regularization constraint into the training objective. Specifically, we added an entropy regularization term to the loss function of our DPC-based iSpikformer (T = 4) trained on the Electricity dataset, encouraging higher entropy in the encoded spike patterns. We varied the regularization weight $\lambda$ from 0.01 to 1.0. This intervention successfully increased the spike train entropy from 2.388 to 3.028; however, the overall model performance remained nearly unchanged.

We hypothesize that this is because the DPC framework already promotes sufficient spike pattern diversity through its temporal embedding and temporal feedback mechanisms. Additional entropy maximization may introduce variability without yielding further improvements in task-relevant representations. This observation suggests that while entropy regularization can enhance diversity, its benefit for downstream performance may be limited when architectural components already address this aspect.

Finally, we note that DPC is designed as a plug-and-play module that replaces the encoder without requiring changes to the loss function or optimization procedure. We therefore conclude that DPC offers a more practical and robust solution compared to modifying the objective function, particularly for general applicability across diverse architectures.

## H  ENERGY CONSUMPTION ANALYSIS

Our energy analysis follows the standard methodology widely adopted in prior SNN studies (Li et al., 2025; Kundu et al., 2021; Su et al., 2024): total per-inference energy is estimated by counting multiply–accumulate (MAC) and accumulate-only (AC) operations and weighting them by their respective energy costs (Horowitz, 2014). The energy computation, considering MAC and AC op-

Table A14: Results of spike shuffling test to quantify the temporal information embedded in encoded spike trains. Clean Acc. and Shuffled Acc. represent the classification accuracy of models before and after applying spike shuffling, respectively. $\Delta$ Acc. represents the difference between them. A larger $\Delta$ Acc. indicates that the temporal information has been significantly disrupted.

| Encoding | $T$ | Clean Acc. (%) | Shuffled Acc. (%) | $\Delta$ Acc. |
|---|---|---|---|---|
| DC | 2 | 76.99 | 76.44 | 0.55 |
|  | 4 | 78.96 | 78.90 | 0.06 |
|  | 6 | 79.14 | 78.96 | 0.18 |
| DPC | 2 | 78.24 | 76.96 | 1.28 |
|  | 4 | 79.58 | 78.73 | 0.85 |
|  | 6 | 80.43 | 79.76 | 0.67 |

erations, is defined as follows:

$$E = T(fr * E_{AC} * O_{AC} + E_{MAC} * O_{MAC}) \tag{A17}$$

, where $T$ is the number of timesteps, $fr$ is the firing ratio, $O_{AC}$ and $O_{MAC}$ are the number of AC and MAC operations, $E_{MAC} \simeq 4.6\text{pJ}$, and $E_{AC} \simeq 0.9\text{pJ}$ (Horowitz, 2014).

## I    LATENCY ANALYSIS

We report the average training time of DPC, measured over 5 epochs after training has stabilized in Table A12. In all cases, DPC introduces a slight increase in training time over direct coding (ranging from 0.7% to 7%). Notably, for the GAC model, we observe a larger increase due to the additional cost of learning spatiotemporal attention in the encoder. We also report an inference time comparison of DPC and DC in Table A13. The additional latency introduced by DPC remains consistently below 4%. This comparison highlights the simplicity and the efficiency of DPC in terms of training time, especially considering the benefits it provides.

## J    TEMPORAL INFORMATION ANALYSIS

---

**Algorithm A2** Spike shuffling for a single spike train

---

**Input:** $A \in \{0, 1\}^t$ - a string of length $t$
**Output:** $\hat{A} \in \{0, 1\}^t$ - shuffled string of length $t$
 1: **if** $A$ **is not** *all-zero* **or** *all-one* **then**
 2:    $\hat{A} \leftarrow A$
 3:    **while** $\hat{A} = A$ **do**
 4:       $\hat{A} \leftarrow$ *randomPermute*$(A)$ //randomly get one of the permutations of the string
 5:    **end while**
 6: **end if**
 7: **return** $\hat{A}$

---

To assess whether encoded spike trains contain temporal information, we conduct a shuffling experiment inspired by (Bu et al., 2023). For each trained SNN, spike trains from the encoding layer are randomly permuted in time to disturb their temporal information except all-zero and all-one patterns, which remain unaffected. This shuffling experiment is conducted on both direct-coded and diverse-pattern-coded models, measuring accuracy to assess how much temporal information is encoded in the spike trains of each scheme. Experiments are conducted using SDT-2-512 on CIFAR100 with timesteps of 2, 4, and 6. The pseudo-code for the shuffling algorithm is provided in Algorithm A2, and the results are presented in Table A14. The direct coding shows only a slight drop in accuracy, consistent with prior findings that spike trains under direct coding lack temporal information (Bu et al., 2023). In contrast, DPC exhibited a notable decline, indicating that its spike trains carrying meaningful temporal structure are disrupted by shuffling.

# K LLM USAGE

During manuscript preparation, we employed OpenAI's ChatGPT, a large language model, solely for proofreading and improving the clarity of writing. Our interaction with the LLM was iterative and strictly limited to language refinement. We confirm that the LLM did not contribute to the conception of research ideas, experimental design, data analysis, or the results reported in this paper. All scientific content and claims are entirely the responsibility of the authors.

