# OpenReview forum: "Enhancing the Representational Power of Spiking Neural Networks via Diversified Spike Patterns"
_ICLR.cc/2026/Conference — ICLR 2026 Conference Withdrawn Submission_

### Official Review · Reviewer_u4d4 · 2025-10-26

**Soundness:** 2
**Presentation:** 3
**Contribution:** 2
**Rating:** 4
**Confidence:** 3

**Summary:**

The authors introduce a neural coding scheme for Spiking Neural Networks (SNNs), called diverse-pattern coding (DPC), which aim to alleviate the lack of diversity in the spike pattern. The two main building blocks of DPC are: a learnable-time dependent embedding and a temporal feedback layer. An entropy-based measure of diversity is introduced and shown experimentally that more diversity correlates with better performance. Numerical experiments are reported, showing that DPC improves in accuracy over the Direct Coding (DC) baseline on a variety of tasks, with static and dynamic data. Comparison of the training time and energy consumption between DC and DPC are also reported.

**Strengths:**

**Clarity** :The paper is well-written, with a clearly articulated methodology that is easy to follow.

**Originality**: A key strength is the simplicity of the proposed scheme, enhancing a SNN model with only two new elements.

**Quality**: Improvements in accuracy seem robust across different tasks and architectures.

**Significance**: Achieving an optimal tradeoff between accuracy and computational (and energy) efficiency is a central challenge in SNNs. This work provides a simple yet effective tool to move towards this goal.

**Weaknesses:**

**Lack of bias terms in the SNN baseline**: I agree that the discrete Leaky Integrate-and-Fire (LIF) model is the most prevalent in applications. However, the version presented here—at least in the main text, Eqs. (1)–(3)—appears to lack a trainable bias term (i.e., a vector $b$ such that $I = W_{\text{enc}} X + b$), which is typically standard in LIF architectures. In this sense, the method proposed here can be interpreted as introducing a time-varying bias with a particular parametrization based on the layer weights, namely $b^t = W_{\text{enc}} E^t$. This constrains the bias to lie within the column space of the weight matrices. While this formulation is interesting, it would be important to clarify whether the baseline SNN architecture includes bias terms, and if not, to provide experimental comparisons against a version that does.

**Only one reset mechanism is considered**: For the same discrete LIF dynamics, alternative reset mechanisms exist—such as the reset by subtraction rule, i.e., $V^t = U^t - V_{\text{th}} S^t$. Under this mechanism, the periodicity analysis presented in the paper does not directly apply. To make the analysis more comprehensive and relevant, it would be valuable to include a baseline using this alternative reset rule or, at least, to discuss how the theoretical framework might extend to it.

**Lack of Neuromorphic implementation discussion**: While it is reasonable not to expect experimental results from a neuromorphic implementation, discussing the potential for such implementations would substantially strengthen the paper. The true energy-efficiency advantages of SNNs are unlikely to be realized on GPU-based simulations. For example, parameter quantization is a key factor, as many neuromorphic hardware platforms impose bit-width constraints on parameter representation. Addressing these aspects would enhance the paper’s relevance regarding practical energy-efficiency implications.

**Discussion specific to static data**: The main analysis seems to focus on static inputs. However, when the method is applied to dynamic data, such as time series, the assumption that $I^t$ remains constant no longer holds. It would be helpful to clarify the interpretation of the time encoding, as well as potential limitations of this approach in handling time-dependent inputs.

**Practical impact**: While the increase in accuracy appears consistent, the magnitude of the improvement is not particularly large. Although the baseline accuracy in this setting is already high, it raises a question about the practical significance of the reported gains.

**Questions:**

1. While the diversity in spike patterns is a compelling idea, one might wonder whether it is truly the *lack of diversity* that limits the baseline SNN’s performance, or rather the *inability of the model* to generate a sufficiently rich set of patterns. After all, during training, it is still possible to observe diverse spike configurations (among the few possible ones illustrated in Fig. 1). Clarifying this distinction could help better isolate the source of the observed limitations.

2. Do the authors expect their results to change when including a trainable bias term (either static or time-varying)?
   *(See also the “Weaknesses” section above for related discussion.)*

3. Similarly, how would the results be affected if a **reset-by-subtraction** mechanism were used instead of a reset to a resting potential? This alternative dynamic might interact differently with the proposed analysis and could offer further insights into the generality of the findings.

4. I recommend including information about the **variance or confidence intervals** of the reported results to better assess their statistical significance. (I noticed that some of this information is provided in the appendix, but including a brief summary in the main text would improve readability.)

5. There seems to be a possible **typo in Fig. 1(b)**: the pattern $\langle0000\rangle$ appears with approximately 2 %, while the text suggests it occurs around 60 %. Please clarify or correct this discrepancy.

6. Could the authors clarify the experimental setting in **Section 3.2.2**? If I understand correctly, the threshold is treated as a hyperparameter and not as a trainable quantity. In some SNN models, this parameter is also learned. Since different choices of learnable parameters can affect the results, it would be helpful to state this explicitly early in the paper.

7. Do the authors have any thoughts on the **potential for neuromorphic implementation** of their strategy? How might parameter quantization or hardware constraints (e.g., limited bit precision) impact its applicability or efficiency?

---

### Official Review · Reviewer_zcCk · 2025-10-28

**Soundness:** 3
**Presentation:** 3
**Contribution:** 3
**Rating:** 6
**Confidence:** 4

**Summary:**

This paper identifies that the repetitive nature of direct coding in Spiking Neural Networks (SNNs) creates imbalanced spike patterns, limiting the network's representational power. To solve this, the authors propose Diverse-Pattern Coding (DPC), a lightweight, plug-and-play replacement that uses a Temporal Embedding and a Temporal Feedback layer to diversify spike patterns by introducing temporal dynamics. Extensive experiments across various tasks and architectures show that DPC consistently improves performance with only a marginal increase in computational cost.

**Strengths:**

1. Clear problem identification and analysis. The paper provide the clear identification and rigorous analysis of a subtle but fundamental problem in a widely-used SNN technique. The framing of the issue as one of "spike train pattern imbalance" and the use of spike train entropy to quantify diversity is both novel and insightful. The analysis provide the bridge from the mathematical properties of LIF neurons to the observed pattern imbalance, provides a principled foundation for the work.
2. Simple and generalizable solution. The proposed DPC is an elegant and well-motivated solution where the two components, Temporal Embedding and Temporal Feedback, directly address the problem of input repetition and lack of temporal dynamics. The method's "plug-and-play" nature makes it highly practical and broadly applicable, which is a significant advantage.
3. The paper is well-written and easy to follow.

**Weaknesses:**

1. The paper rightly points out the limitations of direct coding. While the entropy-based analysis is novel, the core idea that direct coding's repetitiveness is a problem has been touched upon by prior work (Qiu et al., 2024). The paper could be slightly improved by more explicitly positioning its contribution not just as identifying the problem, but as providing a more fundamental, quantitative diagnosis (via entropy) and a more general-purpose solution.
2. The paper’s claim of DPC as a general "plug-and-play" module is not fully substantiated, as its compatibility with other major SNN training paradigms remains unexplored. A large body of SNN research—including ANN-to-SNN conversion, efficiency-optimized training algorithms (e.g., online learning, approximate BPTT), and many knowledge distillation techniques—is fundamentally designed around the simpler, static-input dynamics of direct coding. The new, learnable temporal dependencies introduced by DPC's components could complicate the static activation mapping required for conversion or break the assumptions made by efficient training methods. A discussion of these potential integration challenges is crucial for contextualizing DPC's contribution and understanding the true scope of its applicability across the diverse SNN system.

**Questions:**

1. On the Interplay with the Loss Objective: Could you clarify the specific formulation of your loss objective? Is it based on a temporal average of the final layer's output spikes/potentials before applying the cross-entropy loss, as is common in many SNN works (e.g., SDT [1])? How do you anticipate your analysis on spike train diversity would change if a temporally-decoupled objective (e.g., TET [1], TWKD[2]) were used instead? More broadly, what is your perspective on the interplay between the training objective's design and the richness of the temporal representations learned by the SNN? Do you think DPC, by introducing more temporal variation at the input, would be particularly synergistic with such temporally-decoupled objectives?

2. On the Implications for Online Learning: The work elegantly addresses the input-level redundancy of direct coding. This has interesting implications for online learning algorithms [3,4,5], which have been shown to be somehow redundant when applied to direct-coded SNNs, as they can be simplified to learning a simple rate-based representation [6]. How do you foresee DPC impacting this line of work? Since DPC introduces meaningful temporal variation at the input, does it unlock the potential for these online learning methods to learn more complex temporal representations than was previously possible? Conversely, are there still fundamental theoretical limits on what these approximate gradient methods can learn, even with the richer input provided by DPC?

[1] Temporal efficient training of spiking neural network via gradient re-weighting. ICLR 2022.

[2] Efficient Logit-based Knowledge Distillation of Deep Spiking Neural Networks for Full-Range Timestep Deployment. ICML 2025.

[3] A solution to the learning dilemma for recurrent networks of spiking neurons. Nature communications.

[4] Online training through time for spiking neural networks. NeurIPS 2022.

[5] Towards memory-and time-efficient backpropagation for training spiking neural networks. ICCV 2023.

[6] Advancing training efficiency of deep spiking neural networks through rate-based backpropagation. NeurIPS 2024.

---

### Official Review · Reviewer_U6GV · 2025-10-29

**Soundness:** 2
**Presentation:** 3
**Contribution:** 2
**Rating:** 4
**Confidence:** 5

**Summary:**

The authors address a problem arising when encoding real and static values before processing them in a SNN. Direct coding performed on real and static values leads to periodicity in the behavior of the first spiking neurons. The expression power of these neurons is thus heavily dependent on the time constant and the spiking threshold of the neurons, with large regions of zero-spikes or spike-only regimes. This work first proposes a study of the effect of spike-train entropy on model performance for static tasks. Then, it introduces a network augmentation called Diverse Pattern Coding (DPC) composed of two modules, the temporal embedding and the temporal feedback modules, which leads to an increase in encoded spike-train entropy and network performance.

**Strengths:**

* While the idea of optimizing the structure of the temporal dimension for better learning in SNNs is not new (https://arxiv.org/abs/2306.17597, https://arxiv.org/abs/2305.13909, https://ojs.aaai.org/index.php/AAAI/article/view/29635), the focus on spike-train entropy to structure the temporal embedding space by learning a temporal modulation of the inputs seems new.

* The boost in accuracy they get is worth the increase in complexity, especially on ImageNet with MS-ResNet-34 (more than 3%!)

**Weaknesses:**

* The described problem of periodicity is true only for static inputs, a task for which SNNs are inherently not fitted.

* The study of the effect of entropy on performance is very interesting; however, the justification of how the temporal embedding module initialization helps to increase it is weak. The paper lacks a formal proof of how the entropy is increased by this initialization.

* A complete ablation study should test whether these independent mechanisms help:
1) temporal feedback
2) temporal embedding, with TE and random initialization

* The use of time series in the benchmarks is highly disturbing, as the presented method is developed for static inputs.

* This paper already noted the limited set of periodic spike patterns with static input:
Castagnetti, A., Pegatoquet, A., & Miramond, B. (2023). Neural information coding for efficient spike-based image denoising. arXiv preprint arXiv:2305.11898.
https://arxiv.org/pdf/2305.11898
It should be cited.

* An alternative to the proposed temporal encoding has been proposed in
https://dl.acm.org/doi/10.5555/3692070.3692856
the idea is simple:
a simple 2d conv is applied to the input image with:
Cin = 3 (RGB channels)
Cout = T
the output is [T, W, H] then at each t timestep t we they feed the image [t, :, :] to the SNN.
It would be interesting to compare this approach to the proposed temporal encoding.

**Questions:**

* Why does it make sense to test this method on time-series, neuromorphic data, and natural language understanding? These tasks present an already meaningful temporal dimension. It seems to help, but it's unclear why (all the theoretical sections assume static inputs)

* Will the authors share their code? I strongly encourage them to do so.

---

### Official Review · Reviewer_JHmY · 2025-11-06

**Soundness:** 2
**Presentation:** 3
**Contribution:** 2
**Rating:** 2
**Confidence:** 5

**Summary:**

This paper investigates the representational limitation of direct coding in SNNs, arguing that repeated constant inputs lead to highly imbalanced and low-entropy spike-train patterns that constrain performance. To address this issue, the authors propose Diverse-Pattern Coding (DPC), a new neural direct coding scheme composed of two modules:

(1) a Temporal Embedding (TE) that introduces learnable time-dependent perturbations to each timestep input, and

(2) a Temporal Feedback (TF) layer that feeds spikes from the previous timestep into the current input to enrich temporal dynamics.

They evaluate DPC on a wide range of benchmarks, including CIFAR-10/100, ImageNet, CIFAR10-DVS, time-series forecasting, and NLP tasks, showing consistent accuracy gains.

**Strengths:**

1. The paper identifies a limitation of direct coding in SNNs (pattern repetition and low diversity).

2. The proposed DPC is architecture-agnostic and can be easily integrated into existing SNN models without structural modification.

3. DPC is validated across multiple modalities (vision, neuromorphic, time-series, and text).

4. Clearer background and method description.

**Weaknesses:**

1. Limited innovation. The proposed TE is inherently the positional embeddings, and the proposed TF is inherently the recurrent feedback connection already explored in existing studies.

2. The connection between spike-train entropy and performance improvement is only empirically demonstrated, not theoretically justified. The paper treats entropy as a diversity metric but never proves its causal relation to performance or information capacity.

3. The assumption of periodic spike generation does not hold in networks with built-in recurrent mechanisms.

4. It is hard to attribute the accuracy gain to “diverse patterns” rather than extra parameteters or simple temporal mixing.

**Questions:**

1. How strong is the causal relationship between spike-train entropy and task performance? Can you provide a formal analysis or mutual-information-based justification?

2. The paper assumes that higher spike-train entropy correlates positively with better representational power and accuracy. However, if this hypothesis holds, why not aim for maximum entropy? That is, ensuring all possible spike-train patterns occur with equal probability? In that case, the spike diversity would be maximal.

3. In Figure 2, the authors vary the decay factor τ and the threshold Vₜₕ individually and observe higher spike-train entropy when they increase. However, Eq. (5) clearly shows that τ and Vₜₕ jointly determine the spike-period boundaries. What happens if both parameters are increased simultaneously? Could their synergistic or compensatory interaction naturally produce more balanced spike-train distributions, potentially achieving the same effect as DPC without introducing additional modules?

4. How sensitive is DPC to the initialization of temporal embeddings and the choice of time step T?

Some important suggestions:

1.As a neural coding method, DPC should be evaluated not only against direct coding or GAC but also against other coding schemes such as temporal coding, phase coding, and burst coding. A comprehensive evaluation should consider multiple dimensions, such as sparsity, temporal credit assignment, firing rates, computational efficiency, etc.

2.For static data, is enhancing spike-train diversity truly necessary? Existing rate coding methods, though limited, may provide sufficient information. Their performance in various dataset architectures best proves this. The proposed DPC appears to be an incremental improvement over direct coding, achieving marginal gains at the cost of extra parameters and computation without fundamentally addressing the mentioned limitations of direct coding.

Moreover, since the encoding-layer weights are trainable, the resulting periodic spike patterns in direct coding could also be viewed as learned temporal representations after sufficient training. So why should these learned periodic patterns be considered a weakness rather than an adaptive feature of the model?

3.In tasks such as time-series forecasting and natural language processing, the input sequences already contain explicit temporal information. Is the TE or TF layer redundant? Because it injects additional time variation unrelated to the actual data semantics. In these cases, I think leveraging the time information inherent embedded in the input data itself is truly meaningful.

---

### Note · Authors · 2025-11-14

I have read and agree with the venue's withdrawal policy on behalf of myself and my co-authors.